# Influence of Brewing Method on the Content of Selected Elements in Yerba Mate (*Ilex paraguarensis*) Infusions

**DOI:** 10.3390/foods12051072

**Published:** 2023-03-02

**Authors:** Jędrzej Proch, Anna Różewska, Aleksandra Orłowska, Przemysław Niedzielski

**Affiliations:** Department of Analytical Chemistry, Faculty of Chemistry, Adam Mickiewicz University, Uniwersytetu Poznańskiego 8, 61-614 Poznań, Poland

**Keywords:** yerba mate (*Ilex paraguariensis*), multielemental analysis, extraction parameters, tap water, ultrasound-assisted extraction, ICP OES

## Abstract

In this paper, the effect of the extraction method on the concentrations of selected elements in yerba mate *(Ilex paraguariensis)* infusions is presented. Seven pure yerba mate samples (without additives) were selected, representing various types and countries of origin. An extensive sample preparation procedure was proposed: ultrasound-assisted extraction using two types of extractants (deionized and tap water) at two different temperatures (room and 80 °C). In parallel, the above extractants and temperatures were carried out for all samples by the classical brewing method (without ultrasound). In addition, microwave-assisted acid mineralization was carried out to determine the total content. All the proposed procedures were thoroughly investigated with certified reference material (tea leaves, INCT–TL–1). For the total content of all the determined elements, acceptable recoveries (80–116%) were obtained. All digests and extracts were analyzed by simultaneous ICP OES. For the first time, it was assessed how tap water extraction affects the percentage of extracted element concentrations.

## 1. Introduction

Yerba mate (*Ilex paraguariensis* A.St.–Hil.) is a subtropical, evergreen tree that grows from 8 to 15 m. The plant is native to and exclusively located in southern Brazil, northern Argentina, Paraguay, and Uruguay [1]. Dried leaves, twigs, and stems of *Ilex paraguariensis* are used to prepare a beverage. Yerba mate infusions are consumed widely in South America and approx. 30% of the inhabitants on this continent drink more than 1 L daily [2]. Therefore, it is one of the most commercially cultivated plants [3]. Recently, yerba mate consumption has been expanding beyond South America to Europe, Asia, North America, and Australia [4]. Its popularity reached Poland, which is noticeable in the 8-fold increased import of the raw material in 2012–2018 [2].

Due to the ratio of leaves to twigs, there are two types of yerba mate. The first kind, called “elaborada con palo”, consists of 70% leaves and 30% twigs. The second kind, called “despalada” (or rarely “elaborada sin palo”), consists of 90% leaves and 10% twigs [5]. Moreover, two different types can be distinguished according to the different pre-processing procedures used [6]. For example, green mate are obtained by brief heating (i.e., blanching) at a temperature of 300–500 °C and drying. Alternatively, a roasting process (at 120 °C) can be applied to dried green mate in 15 min [7]. The leaves are generally responsible for the taste and content of the active substance, while other additives (e.g., dried fruit or flower petals) can be added to improve the taste and aroma of the infusion. However, the conditions of processing procedures (i.e., blanching, roasting, drying) strongly depend on the producer and affect the taste as well as the content of various bioactive compounds in yerba mate [1,8], including nutritional quality [9].

Traditionally, vessels (gourd, matero) made of wood, calabash, or porcelain, are used to prepare yerba mate infusions. Dry twigs and leaves (usually 50 g) are put into the vessel and poured over using hot water (65–80 °C). The drink should be left for a few minutes to infuse. After that, it is often drunk using a metal straw called a “bombilla”. Once prepared, yerba mate dregs are brewed several times [10]. This preparation (hot water infusion) can be called “chimarrão” (in Brazil) or “mate” (e.g., in Argentina and Uruguay). Mostly in summer, cold water infusion (“tereré”) is also popular [11].

Apart from numerous chemical compounds such as purine alkaloids, flavonoids, terpenes, polyphenols, and nutrients [2], yerba mate contains many essential trace elements (e.g., Mn, Fe, Zn, Cu) and potentially toxic elements (e.g., As, Ni, Pb, Cd). Their content may depend on various factors, e.g., soil type, harvest season [12], and preprocessing [9]. Moreover, the extractable content of selected essential trace and potentially toxic elements may be different in the case of infusion prepared from green and roasted yerba mate [6]. The total content of elements (in leaves and stems) is not equivalent to the extractable content (in the infusion), a finding that was reported several times [13,14,15]. Therefore, the determination of these elements is important in evaluating the nutritional potential of the plants as well as in providing a risk assessment for their use; however, obtaining the total content is not sufficient in this field.

The temperature of the water used to brew yerba mate may cause differences in the content of elements that were reported before [16,17,18]. The concentrations of all investigated elements were slightly higher in infusions prepared with hot (70–75 °C) rather than cold water (at room temperature) [17]. Other studies indicated again that the water temperature influences the content of elements in the infusions; however, no difference in iron content was detected in yerba mate with origins in Argentina [18]. It was also found out that the extraction of trace elements was disproportionate to the temperature used, and the weak bonding of ions with the material was given as an explanation [16]. However, all the above studies did not elaborate this issue sufficiently. Diverging opinions on the influence of water temperature encourage further research. In addition, many authors used boiling water (at 100 °C) [14,15,19] instead of hot water (65–80 °C), which is incorrect from the consumer point of view.

When approaching yerba mate as a consumer, it is interesting to check the influence of tap water on the extractable content. It is so surprising that this approach has not been reported so far. It was reported that the pH of the extractant as well as the extraction temperature have an impact on the extractable content [16]. In our previous studies, it was presented that 1 M phosphoric acid allowed us to obtain different contents than deionized water allowed (at both lower and higher temperatures) [5]. For this reason, we decided to compare the effect of tap water and deionized water (both hot and cold) on yerba mate infusion.

Evaluating the effect of the extraction method and its parameters in the preparation of yerba mate *(Ilex paraguariensis)* infusions, the content of 23 elements was investigated. Seven yerba mate samples were selected, representing various types and countries of origin. Four procedures of ultrasound-assisted extraction were proposed: two types of extractants (deionized and tap water) at two different temperatures (room and 80 °C). All the above procedures were compared with the classical brewing method (without ultrasound). In addition, microwave-assisted wet mineralization was performed to determine the total content. The proposed procedures of mineralization and extraction were thoroughly tested with certified reference material (tea leaves, INCT–TL–1). All measurements were carried out using inductively coupled plasma optical emission spectrometry (ICP OES). The results were subjected to statistical analysis and discussed with reference to the literature. For the first time, it was assessed how tap water extraction affects the extracted content of elements in yerba mate infusion.

## 2. Materials and Methods

### 2.1. Sampling

Seven products of five brands were bought online from a legal Polish distributor. samples 2 and 7 were of the same brand as well as samples 3 and 4 (Table 1). All the samples were obtained only from 500 g packs (both paper and plastic). The samples were selected to represent different countries (Argentina, Brazil, and Paraguay) and kinds of yerba mate, “con palo” (which consists of up to 30% twigs and at least 70% leaves) and “despalada” (which consists of up to 10% twigs and at least 90% leaves). All the products were pure (100%) and roasted yerba mate (except for sample 2). Three products were repacked and sold as a Polish brand. The full details are shown in Table 1.

### 2.2. Gases and Reagents

Argon (N–5.0, purity 99.999%) was used as a working gas and purchased from Linde Gaz Polska (Kraków, Poland). Standard calibration solutions were prepared from commercially available ICP standards (Romil, Cambridge, UK). An amount of 65% nitric acid (HNO_3_) was obtained from Merck (Darmstadt, Germany).

Deionized (DI) water (≥18 MΩ cm resistivity) was supplied by a Milli-Q water purification system (Merck Millipore, Darmstadt, Germany), and tap water was taken from the laboratory tap (pH: 7.35 ± 0.01, electrical conductivity at 25 °C: 485 ± 5 µS cm^−1^). A list of the elements in the tap water is presented in Appendix A. The water was heated up to 80 °C in the electric kettle Styline TWK8613P (Bosch GmbH, Stuttgart, Germany) after being boiled in 1% (*w*/*v*) citric acid (Merck).

### 2.3. Sample Preparation

#### 2.3.1. Water Extraction

Each sample was homogenized using agate laboratory grinder (Pulverisette, Fritsch GmbH, Idar-Oberstein, Germany). The following procedure (ultrasound-assisted extraction) was conducted similarly to our previous studies [5]. Dry samples were accurately weighted (1.00 ± 0.01 g) and put into a polypropylene test tube. Then, 10.0 mL of either of the following extractants was added: (A) tap water at room temperature (RT), (B) DI water at RT, (C) tap water at 80 °C, (D) DI water at 80 °C, and the extraction assisted by ultrasound was conducted for 30 min at ambient temperature. The samples were subsequently filtered through paper filters, which had been previously washed with 200 mL DI water. The sample solutions were filled up with DI water up to 15 mL. As a reference method, conventional brewing (the same procedure of extraction without ultrasonication) was used for each sample and extractant. Blank samples were also prepared for all the described procedures.

#### 2.3.2. Wet Mineralization

Additionally, dry homogenized samples were accurately weighted (0.300 ± 0.001 g) and mineralized with 65% HNO_3_ (7.0 mL) in closed PTFE containers (55 mL) using the microwave digestion system, Mars 6 (CEM, Matthews, NC, USA). The process was performed in three stages: (1) ramping the temperature (20 min), (2) maintaining it at 180 °C (20 min), and (3) cooling (20 min). After mineralization, the sample solutions were diluted up to 15 mL with DI water. All digests and extracts were analyzed on the same day using ICP OES.

### 2.4. Instrumentation

An inductively coupled plasma optical emission spectrometer, Agilent 5110 ICP–OES (Santa Clara, CA, USA), was used to determine 23 elements in acid digests (total content) and extracts (extractable content). The typical conditions were applied: radio frequency (RF) power 1.2 kW, auxiliary gas flow 1.0 L min^−1^, nebulizer gas flow 0.7 L min^−1^, plasma gas flow 12.0 L min^−1^, and axial plasma view. The echelle grating optics were thermostated (at 35 °C), and the CCD detector (VistaChip II) was cooled using a triple Peltier system (up to −40 °C). The signal was measured in 3 replicates (5 s each). Detection limits (DLs) were ascertained as the three standard deviations (SD) of the multiple blank measurements (n = 10) and the method quantification limits (QLs, including sample preparation). All QLs and emission lines (nm) of the determined elements are presented in Appendix A. All samples were repeated twice (n = 2), and the results were corrected with a blank sample. The analysis of the quality control was performed with five certified reference materials (CRMs), including hardwood biomass material (NIST SRM 2790), tobacco leaves (INCT–OBTL–5, INCT–PVTL–6), and powdered mushrooms (WEPAL–IPE–120, INCT–CS–M–3), and acceptable recoveries (80–120%) were achieved for most of the elements (Appendix A). Additionally, tea leaves (INCT–TL–1) were used to evaluate all the sample preparation procedures (both acid mineralization and water extractions, A–D), comparing water-extractable content with yerba mate. The propagated uncertainty for the whole analytical process (including sample preparation) was estimated to be below 20% (a coverage factor k = 2 for approximately 95% confidence).

### 2.5. Statistical Analysis

The statistical analysis was carried out using the computer program Statistica 13.3 (StatSoft, TIBCO Software Inc., Palo Alto, CA, USA). The data distribution was determined by the Shapiro–Wilk, Lilliefors, and Kolmogorov–Smirnov tests. The normality of the distribution was rejected for most of elements (except Ni and Rb); therefore, the Spearman’s rank correlation coefficient, which is a nonparametric test, was used. For all statistical tests, the probability value *p* = 0.05 was applied.

## 3. Results and Discussion

### 3.1. Microwave-Assisted Acid Mineralization (Total Content)

A total of 18 elements were determined above the quantification limit (AQL) in all the investigated samples (Table 2). Co, Cr, and Pb were found below the quantification limit (BQL) in samples 1, 2, and 4, respectively. In turn, Hg and Mo were found BQL in all the investigated samples. The 18 elements were arranged in the following order (the order of their median values): K > Ca > Mg > Mn > P > S > Al > Fe > Na > Zn > Rb > Sr > Cu > Ni > Se > As > V > Cd. Moreover, the order K > Ca > Mg > Mn > P > S > Al > Fe > the other elements was observed for most of samples (except sample 1, where S > P). This observation was also reported in several studies [13,14,15,17,20]. In the case of the other elements (ranging 20–100 mg kg^−1^), the following orders were generally observed: Na > Zn and Rb > Sr (sample 1, 2, 6, 7). It is worth noting that only the green yerba mate (sample 2) was an exceptional case (in the whole series), where the orders Rb > Zn and Ni > Se were observed. In turn, a different pair (Zn > Na and Sr > Rb) was noticed for samples 3 and 4, which were the same brand. Additionally, sample 5 (Argentinian con palo mate) was an exception for which the following series was noted: Na > Zn and Sr > Rb. Nevertheless, other characteristic orders (in the total content) were not observed when comparing the type and country of origin, which was the case for too few samples (n = 7). It is worth mentioning that higher sulfur content was found in samples 1, 2, and 7; while all the samples were Polish-brand products sold in plastic bags, it is difficult to clearly determine the reason for this.

The total content was generally similar to that found in the latest literature to date (except As and Se, the contents of which were generally lower) [11,13,14,15]. However, lower contents of both As and Se were found in comparison with our previous study [5]. It was recognized that dust residue deposition on foliar tissue may have caused significant excess levels of the selected elements (e.g., As, Fe, Pb, Mo, and V) [21].

The maximum limits of As, Cd, and Pb in yerba mate (as total concentrations) were established within South American legislation (0.6, 0.4, and 0.6 mg kg^−1^, respectively) [22]. The maximum limit for As was slightly exceeded in five samples (0.64–0.72 mg kg^−1^), and acceptable content was found in samples 3 and 5 only. In turn, the amount of Cd slightly exceeded the limit in sample 4 (0.440 mg kg^−1^), and Pb was found AQL in two samples in amounts that were well below the established limit. Except for the case of As, the results are in accordance to the literature data. Moreover, many authors reported that Cd and Pb exceeded the recommended level [3,17,20,21,23] and suggested a revision of this limit [15,24,25].

### 3.2. Ultrasound-Assisted Water Extraction

In the case of Ca, Na, and S in which the concentration of tap water was the highest (Appendix A), no differences were observed between the content of the blank sample and that of those extracts (regardless of the temperature). Tap water extraction is novel, making these results difficult to interpret unambiguously in this study. Nevertheless, the observation seems promising in the context of further research on the influence of tap water on the extractability (leachability) of elements from yerba mate products, and we decided to note it (Table 3). According to this, the content of these elements (Ca, Na, S) were reported as blank equivalent concentrations (BEC, mg kg^−1^).

Ten elements (Al, As, Cu, K, Mg, Mn, Ni, P, Rb, and Zn) were extracted AQL in all samples (n = 7), regardless of the type of water and the temperature used (methods A–D). Apart from Hg and Mo (not determined AQL in the digested samples), Pb and V were also found BQL regardless of the extraction method. In addition, Cd, Fe, and Se were non-extractable in cold tap water (method A). For all samples (n = 7) and methods (A–D), the extraction percentages (as median) were put in ascending order: Fe (0.9%), Cd (4.2%), Ca (11%), Sr (11%), Al (11%), As (24%), Mn (30%), Zn (32%), S (36%), Cu (39%), Mg (42%), Se (46%), P (50%), Ni (55%), K (57%), Co (57%), Na (63%), Cr (67%), and Rb (77%). Cold and hot tap water extracted Sr occasionally (samples 5 and 7), however it was found AQL in all samples if DI water was used. In turn, Se was found to be extracted by DI water AQL in four samples only, a sample in cold DI water (sample 5) and three in hot DI water samples (samples 5–7). However, there were no significant differences between the percentages of extractable content regardless of the procedure.

Comparing the medians of the extraction percentages between procedures A–D, slight differences might be reported. The following orders, representing extractable content gain in high temperatures, were observed for Al, Mg, and Ni (D ≥ C ≥ B ≥ A) and Cd (C ≥ D ≥ B ≥ A). In turn, the gain in deionized water was reported for Co, K, and Sr (D ≥ B ≥ C ≥ A), Cr, Fe (B ≥ D ≥ C ≥ A), and Cu, and P (D ≥ B ≥ A ≥ C). Moreover, cold tap water (procedure A) extracted significantly less Cd, Co, and Cr content than the water used in other procedures (B–D). In the case of elements extracted with DI water only, the order B > D was observed for Na and Se (the gain in low temperature), while the order D > B was observed for Ca and S (the gain in high temperature). On the one hand, the ambiguous orders were found for Mn, Rb, and Zn (D ≥ A ≥ C ≥ B) and As (B ≥ C ≥ A ≥ D). On the other hand, no difference or slight differences in the median percentages were observed between cold and hot tap water for Mn, Rb, and Zn, while this difference was 10% (Rb) in the case of deionized water.

No significant differences were observed between the samples of different countries of origin, types, and packing material when comparing the medians of the extraction percentages. It is surprising that samples 2 and 7, which were both Brazilian mate (the despalada type) of the same brand, showed noticeable differences in the extractable content of various elements. Higher extraction percentages for Fe, Cu, and Zn were observed for sample 2 (green mate) had, while higher percentages of K, Mg, Na, Ni, P, Rb, and S were observed for sample 7 (roasted mate). Moreover, Co and Se were detected AQL in the extracts of sample 7. Similar percentages were found in the case of Al, As, Ca, Mn, and Sr, while Cr and Cd were not extractable in both samples. The explanation for this is probably that a thermal degradation of polyphenols occurred during the preprocesses (i.e., drying and roasting). The influence on the extraction percentages is reported below and suggests that these elements (in which the content increased) were bonded to the polyphenols [6].

On the one hand, our results for seven representative samples suggest that tap water usually does not leach elements as effectively as deionized water does. On the other hand, the research on the influence of tap water extraction on yerba mate infusions is novel, and there is a lack of date in the literature. Nevertheless, it would be impossible to compare the results of leaching with tap water obtained in different laboratories from different countries. Therefore, it cannot be considered as being a scientific method. The effect of temperature on increasing the content of extractable elements is more noticeable for deionized water, a finding which was reported in the literature [17]. However, it cannot be omitted that the selected elements were extracted better in cold deionized water (As, Fe, Na, Se). It was discovered before that the amount of extractable content is not proportional to the temperature used, and the weak bonding of element ions to the material is given as an explanation [16]. The above observations suggest the need to continue research on large series of samples comparing again cold and hot extraction in tap water (or other types of drinking water).

### 3.3. Accuracy of Mineralization and Extraction Procedures

In addition to the five analysis CRMs (wood, leaves, mushrooms), a comprehensive investigation of the applied methods (acid mineralization and water extraction) was carried out using the most similar CRM to yerba mate (tea leaves, INCT–TL–1). The full set of data are shown in Table 4.

Using INCT–TL–1, the method accuracy was estimated for the total content (microwave-assisted acid mineralization) of 22 elements (without Mo, the noncertified value). In this case, acceptable recovery was found for all elements (80–116%). However, there is a lack of a certified value for water-extractable content in the case of INCT–TL–1. Therefore, the extraction percentage of the detected total content was reported instead of the recovery. Five elements (Cd, Hg, Mo, Se, V) were determined as being below the quantification limit (BQL) in the tea leaves extracts regardless of the procedure (A–D). The content of Ca, Na, and S was reported at the level of the blank equivalent concentration (BEC, mg kg^−1^), similarly to the tap water extraction of the yerba mate samples. In turn, Sr was unextractable using tap water (A, C). Other elements were most extractable in hot deionized water (procedure D), except As, which was found BQL. Nevertheless, As was the most extractable element in the other procedures (61–73%), while Fe was the least extractable element overall (0.3–0.6%). It is also noteworthy that Pb was 2–2.5-fold more extractable in procedure D (14%) than in procedures A–C (5.3–6.7%), while Cu was 2.5–3-fold more extractable in procedures B–D (10–13%) than in procedure A (4.0%).

Similarly to the yerba mate samples (n = 7), the medians from all the procedures (A–D) were calculated, arranging the extraction percentages in the following order: Fe (0.5%), Pb (6.6%), Ca (8.5%), Sr (10%), Cu (11%), Mn (13%), Zn (14%), Co (15%), Ni, Mg (18%), S (19%), Cr (23%), Al (25%), P (28%), K (42%), Rb (54%), As (67%), Na (73%). According to this, varying extractability was observed for 23 elements in the tea leaves and yerba mate. The main difference was noted for Cd, Pb, and Se, which were not found BQL in all the extracts of the tea leaves (Cd, Se) and yerba mate (Pb). Additionally, the order of the most extractable three elements was slightly different: Na > As > Rb (tea leaves) and Rb > Cr > Na (yerba mate); however, the percentages of total content (as medians) were higher for the yerba mate (63–77%) than for the tea leaves (54–73%). Moreover, higher extraction percentages in yerba mate were found for Rb, Cr, Co, K, Ni, P, Mg, Cu, S, Zn, Mn, Sr, and Ca, while higher extraction percentages were found for Na, As, and Al in the case of the tea leaves.

The highest similarity in both materials was observed for Sr (10 and 11% of extractable content), while Ca was slightly more extractable from the yerba mate (11%) than from the tea leaves (8.5%). It is worth noting that Hg, Mo, and V were determined BQL in the extracts of both materials regardless of the procedure (A–D). In both materials, As was less extractable in procedure D than in others. Moreover, Fe was the least extractable element in both materials, indicating that Fe predominates in a non-extractable form. In the case of yerba mate, it was reported before that small extractable amounts are probably ionic forms, such as both Fe(II) and Fe(III), coming from dust residue deposition [5,21]. Therefore, a higher percentage (as median) was found for the yerba mate samples (0.9%) than for the tea leaves, INCT–TL–1 (0.5%).

The above results indicate that all 23 elements were extracted differently from both materials (tea leaves and yerba mate). While several CRMs have been identified for tea leaves, e.g., tea leaves (INCT–TL–1, Poland), green tea leaves (NIST SRM 3254, USA), and Chinese green tea (AN–BM02, Czech Republic), no reference material for yerba mate (*Ilex paraguariensis)* has been identified so far. For this reason, an attempt to develop a reference material based on yerba mate seems to be extremely desirable from a scientific point of view.

### 3.4. Ultrasound-Assisted Extraction in Comparison to a Conventional Brewing

Traditionally, one portion of dried yerba mate (approx. 50 g) is poured several times for consumption purposes. However, a procedure of three-stage infusion preparation was investigated before as well as assessing the content of elements in each infusion [13]. From the point of view of sample preparation (especially of a large series), this procedure is laborious and time-consuming. Therefore, an ultrasound-assisted water extraction (USN) procedure was proposed in accordance with a similar scheme that was previously presented [5]. In order to evaluate this method, the same procedure of extraction without ultrasonication (single-step conventional brewing) was applied in parallel for each sample and procedure (A–D). The ultrasound-assisted extractable content was expressed as a relative percentage (%) to the conventional brewing extractable content (considered 100%). This percentage was calculated only if the element was determined AQL in the same sample. These relative percentages were reported as the medians and the range (min–max) for each procedure (A–D) separately (Appendix A). Due to the fact that the extractable content of Ca, Na, and S was at the level of the blank equivalent concentration (BEC) for procedures A and C (tap water extraction), only procedures B and D were compared.

Generally, ultrasound-assisted extraction (USN) allowed the obtainment of higher content than single-step conventional brewing (CON) regardless of the procedure used (A–D). Few elements (Cd, Fe, and Se) were much more extractable with ultrasound assistance, however all of them were unextractable in procedure A (tap water, RT). In turn, Sr was difficult to extract using tap water regardless of the temperature and extraction type (USN or CON). Apart from the elements mentioned above, the others were determined within the same sample whether USN or CON was used. Considering the range of relative percentages (as medians), the advantage of USN over CON was noticeable for procedures A (115–152%) and D (99–199%). Significant differences (if the ratio USN to CON was more than 120%) were observed for 8 and 16 elements for procedures A and D, respectively. In turn, the advantage of USN over CON was unequivocal for procedures B (97–237%) and C (43–125%). It is worth noting that the advantages of CON over USN were observed for selected elements. On one hand, Mg, Cu, P, and Sr were similarly extractable regardless of the extraction type in the case of procedure B. On the other hand, using USN in procedure C allowed the extraction of only 43% of the Cd and 69% of the Sr extracted using CON. The advantage of using CON over USN could be considered significant for these elements; however, this advantage was only found for single samples (no. 4 for Cd and no. 7 for Sr). Other significant differences (if the ratio of USN to CON was more than 120%) were observed only for two elements in procedures B (As, Na) and C (Mn, Cr).

In the case of Cd, the advantage of using CON over USN in hot tap water seems worthy of attention. On one hand, Cd was found AQL in one sample only (Paraguayan con palo) using both USN and CON. On the other hand, many authors reported overestimated Cd levels in many products of yerba mate [3,15,17,20,21,23,24,25]. It was reported that as much as 21% of the tested samples may exceed the recommended Cd levels (0.4 mg kg^−1^) [25]; therefore, it must be pointed out that Cd may be easily extracted under hot tap water brewing. Nevertheless, there is a need to conduct further research in this field.

The above results may be easily compared with those of the hot water extraction procedures reported by Baran et al. (2018) [13]. It was reported that the highest extraction percentage was found in the first infusion (maximally 2.2-fold and 5.1-fold higher than in the second and third infusion, respectively) [13]. According to this, USN with DI water at 80 °C may simulate two-step extraction conventional brewing. It is noteworthy that extraction time can affect the content extracted from yerba mate in the brew. In this study, the extraction time was relatively long (30 min), whereas Baran et al. proposed an extraction time of 3 min for each infusion. Therefore, the extraction percentages were generally higher when using USN than when using three-step brewing [13]. Accordingly, the next step should be to optimize the brewing time, changing this method to reflect the natural conditions without affecting the ease of sample preparation. Nevertheless, the proposed procedure (ultrasound-assisted extraction) may be evaluated as a fast and efficient extraction method reflecting at least double the brewing of yerba mate.

### 3.5. Extraction Percentages in Comparison to the Latest Literature

Extraction with tap water (both cold and hot) is a scientific novelty; therefore, the obtained results cannot be related to other studies. Accordingly, the comparison of the extractable content of investigated elements was based solely on the procedure of extraction with deionized water. Extraction using deionized water at a temperature ranging 70–100 °C is commonly used to prepare yerba mate infusion, and it was previously studied by several authors [13,14,15,16,19]. Comparing the procedure of hot water extraction conducted here to that conducted in the literature (Table 5), both higher and lower percentages were achieved in this study. For most elements, our median percentages are in the middle of the literature range except Cr and Se (higher percentages) and P (lower percentages). According to the literature, Hg content was found BQL in extracts. Contrary to this study, Mo, Pb, and V were detected AQLs by several authors [14,15,17,19]. The comparison of extraction using both hot and cold (at the room temperature) deionized water was conducted before [17], however these procedures are rarely compared in the literature. On the one hand, authors reported slightly a higher content of all the elements in hot water infusions. On the other hand, the reported percentages were the lowest in cold water infusions for most of the elements with a few exceptions (Ca, Fe, Na, V) [17].

The extraction percentage may be associated with many factors, e.g., plant abundance, compounds containing the element, the nature of the chemical element, and the type of extractant (including its temperature and pH) [15]. However, it should be noted that the percentage of extractable content is strongly influenced by the sample preparation procedure (both digests and extracts). For sample decomposition, other authors used dry digestion [13,19], hot plate mineralization [17], heating block mineralization [14,20,26,27], and microwave-assisted wet mineralization [3,5,11,15,24], which is increasingly used. However, there is still a lack of standardization in preparing infusions, and most of the authors proposed their own procedures for yerba mate brewing.

For the consumption of yerba mate infusions, the most common method of infusion is brewing it in a gourd using hot water (70–80 °C). The gourd content is able to be brewed several times with hot as well as cold water [2]. Therefore, the determination of the content of elements in hot and cold infusions simultaneously is significant in extractability studies and risk assessments [17]. However, in the literature, there is a dominance of yerba mate brewing using boiling water (100 °C) [14,15,19], which is not recommended for yerba mate brewing. In the literature, it is suggested that the only risk factor is the ingestion of yerba mate infusions (above 60 °C) [24]. A cancer risk could be associated with certain substances which are present in yerba mate infusions. The carcinogenic potential or the exacerbation of disease effects may be exhibited when large volumes of hot infusions (“chimarrão”) are frequently consumed [28]. However, the daily consumption of yerba mate was not found to pose essential non-carcinogenic human health risks [13]. It was reported that yerba mate drinking (as chimarrão) is safe in an amount of up to 50 g day^−1^ [15].

We suspected that deionized boiling water (at 100 °C) may extract a higher content of investigated elements than deionized hot water (at 80 °C) can. Therefore, both boiling and hot deionized water was used in the ultrasound-assisted extraction of three yerba mate samples to evaluate this difference. The results are reported (Appendix A) as relative percentages (assuming that of procedure D is 100%). All elements that were determined in the infusions were more extractable in boiling water (as median) in the range from 101% (Ca) to 169% (Rb) of the content extracted in hot water (80 °C). Definitely insignificant differences were only observed for Ca and Cd (101% and 103% as medians, respectively); however, the difference was at least 13% for the other elements in boiling water. Therefore, the use of boiling water for the sample preparation of yerba mate infusion seems inappropriate, and revisions of the concentrations are suggested.

### 3.6. Spearman’s Rank Correlation Test

Due to the normality of the data distribution being rejected as well as the small number of samples (n = 7), the Spearman’s rank correlation coefficient (r_s_) was applied (Figure 1), presenting the pairwise associations between the total content and the extraction procedures (A–D). Furthermore, only 10 elements were selected (the contents of which were determined AQL for each digest and extract).

Each procedure (A–D) was associated with the total content. For procedure A (cold tap water extraction), there was a pair almost fully positively correlated (r_s_ ≥ 0.9): Rb(total)/Rb(ext). Moreover, strong positive correlations (r_s_ ≥ 0.7) were observed between Ni(total)/Ni(ext), Ni(total)/Mn(ext), Ni(total)/Mg(ext), Cu(total)/Mg(ext), and Zn(total)/Zn(ext). In turn, strong negative correlations (r_s_ ≤ −0.7) were observed between As(total)/As(ext), K(total)/Mg(ext), and Zn(total)/Rb(ext). Other positive and negative correlations were not statistically significant (*p* < 0.05). For procedure B (cold deionized water extraction), there were two pairs almost fully positively correlated (r_s_ ≥ 0.9): Cu(total)/Mg(ext) and Ni(total)/Ni(ext). Additionally, strong positive correlations (r_s_ ≥ 0.7) were observed between P(total)/P(ext), Zn(total)/Zn(ext), Mg(total)/Mg(ext), Zn(total)/Cu(ext), Zn(total)/K(ext), Ni(total)/Mn(ext), and Mn(total)/Rb(ext). A strong negative correlation (r_s_ ≤ −0.7) was observed only between K(total)/Mg(ext). Other positive and negative correlations were not statistically significant (*p* < 0.05). For procedure C (hot tap water extraction), there was a pair almost fully positively correlated (r_s_ ≥ 0.9): Ni(total)/Ni(ext). Moreover, strong positive correlations (r_s_ ≥ 0.7) were observed between P(total)/P(ext), Zn(total)/Zn(ext), Ni(total)/Mn(ext), Cu(total)/Mg(ext), and Mg(total)/Mg(ext). A strong negative correlation (r_s_ ≤ −0.7) was observed only between Zn(total)/Rb(ext). Other positive and negative correlations were not statistically significant (*p* < 0.05). For procedure D (hot deionized water extraction), there were two pairs almost fully positively correlated (r_s_ ≥ 0.9): Ni(total)/Ni(ext) and Ni(total)/Mn(ext). Additionally, strong positive correlations (r_s_ ≥ 0.7) were observed between Mg(total)/Mg(ext), Mn(total)/Mn(ext), P(total)/P(ext), Zn(total)/Zn(ext), and Cu(total)/Mg(ext). There was also a pair almost fully negatively correlated (r_s_ ≤ −0.9): Zn(total)/Rb(ext). Moreover, strong negative correlations (r_s_ ≤ −0.7) were observed between K(total)/Mg(ext) and K(total)/Al(ext). Other positive and negative correlations were not statistically significant (*p* < 0.05).

The following positive correlations between total and extractable content were observed for all procedures (A–D): Ni(total)/Ni(ext), Ni(total)/Mn(ext), Zn(total)/Zn(ext), and Cu(total)/Mg(ext). Moreover, a strong positive correlation between Mg(total) and Mg(ext) was observed for procedures B–D. The following negative correlations were observed for three procedures: Zn(total)/Rb(ext) (without B) and K(total)/Mg(ext) (without C). It is noteworthy that similar correlations were found in the literature. For the total extractable content and the water-extractable content, an almost fully positive correlation (r_s_ ≥ 0.9) was reported for Mn and Ni, while a strong correlation (r_s_ ≥ 0.7) was reported for K, Mg, P, and Zn [15]. Moreover, strong and moderate (0.4 ≤ r_s_ < 0.7) positive correlations were also observed for Ni/Mn (0.78) and Mn/Mg (0.67), respectively [13]. Similar correlations between the total extractable content and the acid-extractable content were also reported in our previous studies. Moderate positive correlations were observed for the following pairs: Ni(total)/Mn(ext) (0.42) and Mn(total)/Ni(ext) (0.40), while a weak positive correlation was observed for Al(ext)/Ni(total) (0.33) and Cu(ext)/Mn(total) (0.27). There was also an observed weak negative correlation: Mn(ext)/Zn(total) (–0.30). All the above correlations are statistically significant (*p* < 0.05) [5]. It is presumed that frequently reported significant correlations between Ni and Mn may be characteristic of yerba mate.

## 4. Conclusions

The influence of the extraction method and its parameters on the elemental content have been evaluated in this study. By comparing the extractable content of elements between procedures, it was found that the influence of tap water extraction (including its temperature) was difficult to interpret unambiguously (e.g., Ca, Na, S). Moreover, tap water extraction was conducted for the first time in the field; therefore, further studies are needed to understand the unnoticeable extractable content of these elements. The proposed procedures for sample preparation (microwave-assisted mineralization and ultrasound-assisted extraction) were comprehensively tested using certified reference material (tea leaves); however, significant differences in extraction percentages were observed between both sample matrices. Comparing the ultrasound-assisted extraction method and a conventional brewing method, it was found that a higher extractable content was generally obtained using the ultrasound method. Nevertheless, ultrasound-assisted extraction can be successfully used to prepare the infusion (reflecting at least double the brewing of yerba mate). Boiling water has been proven to provide a higher extractable content for all elements, and we recommend not applying this procedure. Although many parameters of the extraction procedure have been evaluated, further studies are needed to be carried out in this field. However, the obtained results suggest that tap water does not leach elements as effectively as deionized water does. We hypothesize that tap water is a less aggressive extractant than deionized water due to the occurrence of bicarbonate buffering (which thus stabilizes the pH).

## Figures and Tables

**Figure 1 foods-12-01072-f001:**
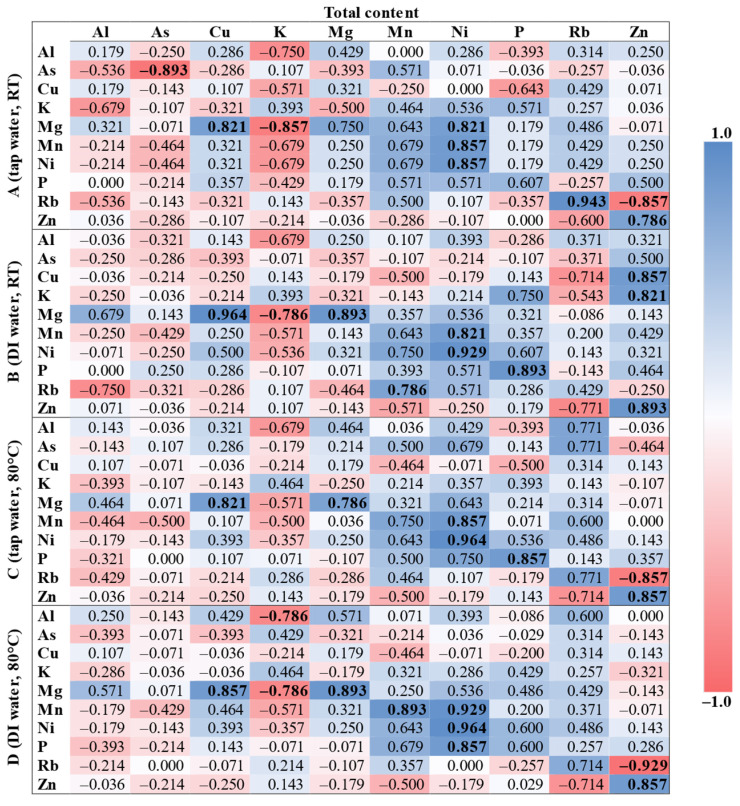
The Spearman’s correlation matrix of 10 elements for total and extractable content. Statistically significant coefficients (*p* < 0.05) are bolded.

**Table 1 foods-12-01072-t001:** Sample characteristics.

Sample No.	Type (Kind)	Country of Origin	Packing Type (Weight)
1	Con palo (roasted)	Paraguay *	plastic pack (500 g)
2 ^a^	Despalada (green)	Brazil *	plastic pack (500 g)
3 ^b^	Despalada (roasted)	Paraguay	paper pack (500 g)
4 ^b^	Con palo (roasted)	Paraguay	paper pack (500 g)
5	Con palo (roasted)	Argentina	paper pack (500 g)
6	Con palo (roasted)	Argentina	paper pack (500 g)
7 ^a^	Despalada (roasted)	Brazil *	plastic pack (500 g)

^a,b^—the same brands; *—products sold as the Polish brand.

**Table 2 foods-12-01072-t002:** Total content (mean ± SD, mg kg^−1^) of 23 elements in yerba mate *(Ilex paraguariensis)*.

	Samples						
Elements	1	2	3	4	5	6	7
Al	398 ± 34	304 ± 16	325 ± 6	375 ± 26	354 ± 37	268 ± 32	372 ± 6
As	0.71 ± 0.07	0.64 ± 0.10	0.56 ± 0.10	0.72 ± 0.09	0.38 ± 0.03	0.71 ± 0.11	0.71 ± 0.02
Ca	9840 ± 1190	7280 ± 252	8910 ± 463	10,200 ± 883	9570 ± 663	8100 ± 281	10,400 ± 360
Cd	0.123 ± 0.002	0.294 ± 0.005	0.381 ± 0.020	0.440 ± 0.023	0.318 ± 0.017	0.270 ± 0.014	0.315 ± 0.003
Co	0.26 ± 0.01	BQL	0.25 ± 0.03	0.24 ± 0.02	0.32 ± 0.03	0.26 ± 0.01	0.20 ± 0.02
Cr	0.84 ± 0.01	BQL	0.24 ± 0.05	0.17 ± 0.01	0.51 ± 0.05	0.22 ± 0.02	BQL
Cu	9.72 ± 0.34	6.57 ± 0.34	8.19 ± 0.57	8.24 ± 1.00	8.38 ± 1.02	7.79 ± 0.13	10.4 ± 0.1
Fe	197 ± 37	108 ± 1	177 ± 6	309 ± 11	208 ± 14	140 ± 5	177 ± 3
Hg	BQL	BQL	BQL	BQL	BQL	BQL	BQL
K	11,400 ± 395	15,500 ± 805	13,300 ± 230	12,700 ± 880	10,500 ± 546	12,300 ± 213	11,500 ± 100
Mg	6120 ± 212	2920 ± 51	4140 ± 143	4300 ± 298	4580 ± 317	3860 ± 201	5450 ± 94
Mn	1160 ± 40	1250 ± 22	1170 ± 41	964 ± 67	1630 ± 141	1510 ± 183	1690 ± 29
Mo	BQL	BQL	BQL	BQL	BQL	BQL	BQL
Na	70.8 ± 6.1	104 ± 4	76.1 ± 1.3	73.3 ± 2.5	75.2 ± 3.9	62.5 ± 0.5	72.3 ± 0.6
Ni	2.98 ± 0.03	0.70 ± 0.02	3.47 ± 0.18	2.94 ± 0.25	3.60 ± 0.37	3.67 ± 0.13	4.70 ± 0.08
P	691 ± 48	882 ± 61	925 ± 48	955 ± 83	904 ± 63	921 ± 80	965 ± 33
Pb	BQL	BQL	BQL	BQL	BQL	0.39 ± 0.02	0.41 ± 0.08
Rb	33.6 ± 2.9	52.9 ± 3.7	24.7 ± 0.9	21.6 ± 2.2	25.1 ± 1.3	34.7 ± 2.4	32.9 ± 0.3
S	711 ± 25	710 ± 37	563 ± 98	529 ± 82	520 ± 72	452 ± 16	642 ± 22
Se	1.02 ± 0.01	0.79 ± 0.07	0.88 ± 0.14	0.90 ± 0.17	1.08 ± 0.15	0.82 ± 0.16	1.14 ± 0.13
Sr	32.5 ± 3.9	20.6 ± 0.4	30.7 ± 2.7	37.6 ± 2.6	29.8 ± 2.1	24.3 ± 1.3	25.0 ± 1.3
V	0.76 ± 0.08	0.31 ± 0.01	0.68 ± 0.05	1.32 ± 0.20	0.89 ± 0.06	0.71 ± 0.14	0.66 ± 0.06
Zn	39.7 ± 0.7	24.5 ± 3.0	77.7 ± 2.7	86.1 ± 3.0	61.4 ± 3.2	55.8 ± 1.0	47.1 ± 0.4

BQL—below (method) quantification limit.

**Table 3 foods-12-01072-t003:** Extraction percentage (% of total content) of selected elements in 7 samples of yerba mate *(Ilex paraguariensis)*.

Method	A (Tap Water, RT)		B (DI Water, RT)		C (Tap Water, 80 °C)		D (DI Water, 80 °C)		All Methods (A–D)	
Elements	Median (Min–Max)	AQL	Median (Min–Max)	AQL	Median (Min–Max)	AQL	Median (Min–Max)	AQL	Median (Min–Max)	AQL
Al	10 (5.7–15)	7	11 (7.4–18)	7	11 (6.4–16)	7	14 (8.6–18)	7	11 (5.7–18)	28
As	24 (20–53)	7	26 (12–60)	7	25 (16–38)	7	21 (15–35)	7	24 (12–60)	28
Ca	BEC	0	10 (7.7–13)	7	BEC	0	11 (7.5–15)	7	11 (7.5–15)	14
Cd	BQL	0	3.2 (2.9–4.4)	3	5.1 (2.7–6.4)	4	4.0 (3.5–5.2)	3	4.2 (2.7–6.4)	10
Co	47 (22–62)	6	58 (40–64)	6	57 (35–66)	6	60 (49–74)	6	57 (22–74)	24
Cr	57 (45–80)	5	68 (38–99)	5	67 (50–99)	5	67 (59–99)	5	67 (38–99)	20
Cu	39 (2.0–43)	7	39 (3.2–43)	7	35 (1.6–46)	7	41 (4.0–48)	7	39 (1.6–48)	28
Fe	BQL	0	1.4 (0.9–2.6)	7	0.5 (0.1–1.1)	7	0.7 (0.2–1.5)	7	0.9 (0.1–2.6)	21
K	54 (39–62)	7	58 (38–60)	7	55 (45–64)	7	60 (50–72)	7	57 (38–72)	28
Mg	39 (28–49)	7	41 (25–48)	7	43 (30–50)	7	47 (36–55)	7	42 (25–55)	28
Mn	30 (21–40)	7	29 (18–36)	7	30 (22–35)	7	33 (25–34)	7	30 (18–40)	28
Na	BEC	0	64 (57–85)	7	BEC	0	62 (40–83)	7	63 (40–85)	14
Ni	53 (20–65)	7	56 (37–61)	7	56 (41–68)	7	57 (41–65)	7	55 (20–68)	28
P	49 (33–58)	7	50 (28–51)	7	47 (36–57)	7	50 (41–64)	7	50 (28–64)	28
Rb	78 (57–99)	7	73 (43–84)	7	77 (55–89)	7	83 (63–99)	7	77 (43–99)	28
S	BEC	0	34 (17–43)	7	BEC	0	37 (21–42)	7	36 (17–43)	14
Se	BQL	0	47*	1	BQL	0	45 (43–56)	3	46 (43–56)	4
Sr	4.8 (3.9–5.6)	2	9.4 (7.5–18)	7	7.0 *	1	12 (7.8–25)	7	11 (3.9–25)	17
Zn	32 (10–38)	7	30 (11–36)	7	32 (11–40)	7	34 (13–41)	7	32 (10–41)	28

RT—room temperature; DI—deionized (water); AQL—above (method) qualification limit (the amount of results exceeding AQL); BQL—below (method) quantification limit (given if AQL = 0); BEC—the content reported as blank equivalent concentration (mg kg^−1^); details in the text; *—the value (given if AQL = 1).

**Table 4 foods-12-01072-t004:** ICP OES analysis of tea leaves (INCT–TL–1), prepared using microwave-assisted mineralization (total content), and four procedures of ultrasound-assisted extraction (A–D). All results presented as mean ± SD (n = 3), except where noted.

	Total Content (Microwave-Assisted Mineralization)			Method A (Tap Water, RT)		Method B (DI Water, RT)		Method C (Tap Water, 80 °C)		Method D (DI Water, 80 °C)	
Element	Certified ± U (mg kg^−1^)	Detected (mg kg^−1^)	Recovery (%)	Detected (mg kg^−1^)	Extracted ^A^ (%)	Detected (mg kg^−1^)	Extracted ^A^ (%)	Detected (mg kg^−1^)	Extracted ^A^ (%)	Detected (mg kg^−1^)	Extracted ^A^ (%)
Al	2290 ± 280	2120 ± 38	92 ± 2	267 ± 21	13 ± 1	497 ± 39	24 ± 2	552 ± 44	26 ± 2	746 ± 59	35 ± 3
As	0.106 ± 0.02	0.18 ± 0.01	110 ± 9	0.071 ± 0.006	61 ± 5	0.078 ± 0.006	67 ± 5	0.085 ± 0.007	73 ± 6	BQL	–
Ca	5820 ± 280	6390 ± 134	110 ± 2	BEC	–	445 ± 56	7.0 ± 0.9	BEC	–	631 ± 79	10 ± 1
Cd	0.030 ± 0.004	0.028 ± 0.002	93 ± 7	BQL	–	BQL	–	BQL	–	BQL	–
Co	0.387 ± 0.042	0.36 ± 0.02	93 ± 6	0.042 ± 0.003	12 ± 1	0.047 ± 0.003	13 ± 1	0.063 ± 0.004	18 ± 1	0.065 ± 0.004	18 ± 1
Cr	1.91 ± 0.22	1.71 ± 0.05	90 ± 2	0.236 ± 0.015	14 ± 1	0.350 ± 0.023	21 ± 1	0.425 ± 0.028	25 ± 2	0.572 ± 0.038	34 ± 2
Cu	20.4 ± 1.5	19.8 ± 0.5	97 ± 2	0.785 ± 0.092	4.0 ± 0.5	1.93 ± 0.23	10 ± 1	2.27 ± 0.27	12 ± 1	2.61 ± 0.31	13 ± 2
Fe	432 ^i^	367 ± 8	85 ± 2	1.07 ± 0.08	0.3	2.04 ± 0.16	0.6 ± 0.1	1.65 ± 0.13	0.4	2.30 ± 0.18	0.6 ± 0.1
Hg	0.005 ± 0.001 *	BQL	–	BQL	–	BQL	–	BQL	–	BQL	–
K	17,000 ± 1200	15,700 ± 345	92 ± 2	3050 ± 194	19 ± 1	6190 ± 395	39 ± 3	6950 ± 444	44 ± 3	8110 ± 518	52 ± 3
Mg	2240 ± 170	1860 ± 11	83 ± 0	124 ± 9	6.7 ± 0.5	281 ± 20	15 ± 1	387 ± 27	21 ± 1	453 ± 32	24 ± 2
Mn	1570 ± 110	1530 ± 15	98 ± 1	136 ± 6	8.8 ± 0.4	157 ± 7	10	242 ± 11	16 ± 1	247 ± 11	16 ± 1
Mo	x	BQL	–	BQL	–	BQL	–	BQL	–	BQL	–
Na	24.7 ± 3.2	23.4 ± 0.8	95 ± 3	BEC	–	11.4 ± 0.5	49 ± 2	BEC	–	22.8 ± 1.0	98 ± 4
Ni	6.12 ± 0.52	4.91 ± 0.03	80 ± 1	0.516 ± 0.028	11 ± 1	0.692 ± 0.04	14 ± 1	1.05 ± 0.06	21 ± 1	1.23 ± 0.07	25 ± 1
P	1800 ^i^	1560 ± 19	87 ± 1	255 ± 19	16 ± 1	402 ± 31	26 ± 2	481 ± 37	31 ± 2	628 ± 48	40 ± 3
Pb	1.78 ± 0.24	1.72 ± 0.15	97 ± 8	0.112 ± 0.005	6.5 ± 0.3	0.092 ± 0.004	5.3 ± 0.3	0.115 ± 0.005	6.7 ± 0.3	0.247 ± 0.012	14 ± 1
Rb	81.5 ± 6.5	75.3 ± 1.9	92 ± 2	20.9 ± 1.1	28 ± 1	35.6 ± 1.9	47 ± 3	46.1 ± 2.4	61 ± 3	57.0 ± 3.0	76 ± 4
S	2470 ± 250	2870 ± 169	116 ± 7	BEC	–	407 ± 43	14 ± 2	BEC	–	657 ± 70	23 ± 2
Se	0.076 ^i^*	BQL	–	BQL	–	BQL	–	BQL	–	BQL	–
Sr	20.8 ± 1.7	18.4 ± 0.4	88 ± 2	BQL	–	1.47 ± 0.16	8.0 ± 0.9	BQL	–	2.24 ± 0.24	12 ± 1
V	1.97 ± 0.37	1.89 ± 0.07	96 ± 4	BQL	–	BQL	–	BQL	–	BQL	–
Zn	34.7 ± 2.7	28.0 ± 0.4	81 ± 1	2.46 ± 0.19	8.8 ± 0.7	3.56 ± 0.28	13 ± 1	4.23 ± 0.33	15 ± 1	5.65 ± 0.44	20 ± 2

RT—room temperature; DI—deionized (water); U—expanded uncertainty of certified value; SD—standard deviation; ^A^—extraction percentage of detected total content; ^i^—informative value; *—value below MQL; x—non-certified value.

**Table 5 foods-12-01072-t005:** Extractable content with deionized water (as % of total content, mean) compared between this study and the literature data.

Ref.	Pozebon et al. 2015 [14]	Barbosa et al. 2015 [19]	Baran et al. 2017 [13]	Olivari et al. 2020 [17]		Ulbrich et al. 2022 [15]	This Study	
Temp (°C)	100	100	85	RT	70–75	100	RT	80
Infusion (ml ÷ g)	20 ÷ 0.5	80 ÷ 10	200 ÷ 25	100 ÷ 10	100 ÷ 10	20 ÷ 0.5	10 ÷ 1	10 ÷ 1
Al	1	18	ND	4.3	5.0	15	12	13
As	48	49	ND	ND	ND	18	27	22
Ca	22	12	0.34	1.8	1.9	17	10	12
Cd	53	55	2	1.8	2.8	13	3.5	4.2
Co	65	86	ND	BQL	BQL	62	54	62
Cr	26	60	9	5.5	7.0	ND	72	73
Cu	42	64	15	9.6	11.1	65	32	37
Fe	15	6	0.13	1.1	1.3	2	1.6	0.8
Hg	ND	ND	ND	ND	ND	ND	BQL	BQL
K	75	80	73	9.3	9.4	92	52	59
Mg	55	74	7	6.4	7.1	67	38	46
Mn	53	28	69	5.5	7.3	53	27	31
Mo	50	57	ND	BQL	5.4	30	BQL	BQL
Na	ND	84	3	6.9	8.0	ND	67	57
Ni	60	88	15	12.2	13.5	90	52	56
P	51	72	ND	ND	ND	57	43	52
Pb	75	44	6	3.3	3.5	17	BQL	BQL
Rb	66	ND	ND	ND	ND	91	68	81
S	ND	ND	ND	ND	ND	59	33	36
Se	ND	ND	ND	ND	ND	24	47 *	48
Sr	2	ND	ND	1.6	1.8	17	11	13
V	37	80	ND	9.8	7.9	1	BQL	BQL
Zn	32	34	8	6.1	6.9	45	27	32

RT—room temperature; ND—no data; BQL—below (method) quantification limit; *—the value (given if n = 1).

## Data Availability

Data is contained within the article.

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
