# Peer review of "Influence of Brewing Method on the Content of Selected Elements in Yerba Mate (Ilex paraguarensis) Infusions"

_foods, 2023, doi:10.3390/foods12051072_

Round 1

Reviewer 1 Report

The manuscript with the title: "Influence of extraction parameters on the content of selected elements in yerba mate (Ilex paraguarensis) infusion" is about the effect of the extraction method on the concentrations of selected elements in yerba mate. 

In general, the manuscript is interesting and can fit the journal's aims and scopes, but needs major improvement in terms of presentation.

1- Title: Revise it to show more related to the journal

2- Abstract: It needs huge revision. For example, it is not common to write in the abstract: "The results were subjected to statistical analysis and discussed with the literature.", Analysis should clearly mention in abstract or remove the sentence.

3- Keywords: Revise the keywords, you mostly write"Key Phrase". Also, choose keywords other than the main words in the title to improve the visibility of the article.

4- Introduction: Update your literature, for example, you reported data from 2012 to 2018, as you know after Covid-19 everything changed and you need to update your data.

5- Materials: Bring all materials in a separate subheading.

6- Results and discussions are OK.

7- The conclusion is too long. Make it short to justify your hypothesis and may recommend future research.

8- References: Try to replace old references by 2020+

Author Response

The manuscript with the title: "Influence of extraction parameters on the content of selected elements in yerba mate (Ilex paraguarensis) infusion" is about the effect of the extraction method on the concentrations of selected elements in yerba mate.

In general, the manuscript is interesting and can fit the journal's aims and scopes, but needs major improvement in terms of presentation.

  1. Title: Revise it to show more related to the journal

The title has been changed.

Influence of brewing method on the content of selected elements in yerba mate (Ilex paraguarensis) infusions

  1. Abstract: It needs huge revision. For example, it is not common to write in the abstract: "The results were subjected to statistical analysis and discussed with the literature.", Analysis should clearly mention in abstract or remove the sentence.

The sentence quoted by the Reviewer has been removed.

  1. Keywords: Revise the keywords, you mostly write "Key Phrase". Also, choose keywords other than the main words in the title to improve the visibility of the article.

We agree with the Reviewer and the keywords has been revised to increase the visibility of the article: extraction parameters, tap water.

  1. Introduction: Update your literature, for example, you reported data from 2012 to 2018, as you know after Covid-19 everything changed and you need to update your data.

The following references has been added in the Introduction.

  • Ribeiro, S.A.O.; da Silva, C.S.; de Araújo Nogueira, A.R.; Garcia, E.E. Solubility of Cd, Cr, Cu, Ni, and Pb and Its Correlation with Total Polyphenols and Soluble Melanoidins in Hot Infusions of Green and Roasted Mate. Biol Trace Elem Res 2022, doi:10.1007/s12011-022-03314-3.
  • Croge, C.P.; Cuquel, F.L.; Pintro, P.T.M. Yerba Mate: Cultivation Systems, Processing and Chemical Composition. A Review. agric. (Piracicaba, Braz.) 2020, 78, doi:10.1590/1678-992X-2019-0259.
  • Ramirez, M.R.; Mohamad, L.; Alarcon-Segovia, L.C.; Rintoul, I. Effect of Processing on the Nutritional Quality of Ilex Paraguariensis. Applied Sciences 2022, 12, 2487, doi:10.3390/app12052487.
  • RzÄ…sa-Duran, E.; Kryczyk-Poprawa, A.; Drabicki, D.; Podkowa, A.; SuÅ‚kowska-Ziaja, K.; Szewczyk, A.; KaÅ‚a, K.; Opoka, W.; ZiÄ™ba, P.; Fidurski, M.; et al. Yerba Mate as a Source of Elements and Bioactive Compounds with Antioxidant Activity. Antioxidants 2022, 11, 371, doi:10.3390/antiox11020371.
  • Camotti Bastos, M.; Cherobim, V.F.; Reissmann, C.B.; Fernandes Kaseker, J.; Gaiad, S. Yerba Mate: Nutrient Levels and Quality of the Beverage Depending on the Harvest Season. Journal of Food Composition and Analysis 2018, 69, 1–6, doi:10.1016/j.jfca.2018.01.019.

  1. Materials: Bring all materials in a separate subheading.

The following subheadings have been added in  the paragraph “2.3. Sample Preparation”: “2.3.1. Water extraction” and “2.3.2. Wet mineralization”. We decided to not divide the paragraph “2.2. Gases and Reagents” because of the paragraph “Gases” would be created for one sentence only.

  1. Results and discussions are OK.

We sincerely thank the Reviewer for the positive reception of this section.

  1. The conclusion is too long. Make it short to justify your hypothesis and may recommend future research.

The conclusion has been shortened according to the Reviewer’s comment.

  1. References: Try to replace old references by 2020+

Compering the obtained results with the literature (Table 5), we chose the latest studies presented multielemental analyses of yerba mate samples, both digests and extracts. Therefore the newest publications (2020 and newer) were Olivari et al. 2020 and Ulbrich et al. 2022 (Table 5). However, the following references have been added in the text:

  • Ribeiro, S.A.O.; da Silva, C.S.; de Araújo Nogueira, A.R.; Garcia, E.E. Solubility of Cd, Cr, Cu, Ni, and Pb and Its Correlation with Total Polyphenols and Soluble Melanoidins in Hot Infusions of Green and Roasted Mate. Biol Trace Elem Res 2022, doi:10.1007/s12011-022-03314-3.
  • Croge, C.P.; Cuquel, F.L.; Pintro, P.T.M. Yerba Mate: Cultivation Systems, Processing and Chemical Composition. A Review. agric. (Piracicaba, Braz.) 2020, 78, doi:10.1590/1678-992X-2019-0259.
  • Ramirez, M.R.; Mohamad, L.; Alarcon-Segovia, L.C.; Rintoul, I. Effect of Processing on the Nutritional Quality of Ilex Paraguariensis. Applied Sciences 2022, 12, 2487, doi:10.3390/app12052487.
  • RzÄ…sa-Duran, E.; Kryczyk-Poprawa, A.; Drabicki, D.; Podkowa, A.; SuÅ‚kowska-Ziaja, K.; Szewczyk, A.; KaÅ‚a, K.; Opoka, W.; ZiÄ™ba, P.; Fidurski, M.; et al. Yerba Mate as a Source of Elements and Bioactive Compounds with Antioxidant Activity. Antioxidants 2022, 11, 371, doi:10.3390/antiox11020371.
  • Camotti Bastos, M.; Cherobim, V.F.; Reissmann, C.B.; Fernandes Kaseker, J.; Gaiad, S. Yerba Mate: Nutrient Levels and Quality of the Beverage Depending on the Harvest Season. Journal of Food Composition and Analysis 2018, 69, 1–6, doi:10.1016/j.jfca.2018.01.019.

Reviewer 2 Report

Dear authors,

I consider your paper entitled "Influence of extraction parameters on the content of selected elements in yerba mate (Ilex paraguarensis) infusions" can be  a minor revision.

In the following there are some observations: 

1. The meaning for AQL, BQL, and BEC should be given, respectively, in line 161, line 162, and line 201, where these abbreviations appear for the first time

2. Lines 207 - 209 – why there are not reported the percentages for Ca, Sr, and K?

3. You say that “the research on the influence of tap water extraction on yerba mate infusions is a novelty and there is a lack of the literature data” (lines 228, 229)

Correlation between the results obtained in different laboratories, from different countries, for leaching with tap water, I don’t think to be possible (it cannot be considered as being a scientific method).  

4. Same notation for two different chapters, i.e. 3.3. for “Accuracy of mineralization and extraction procedures” and for “Ultrasound–assisted extraction in comparison to a conventional brewing”.  

Author Response

I consider your paper entitled "Influence of extraction parameters on the content of selected elements in yerba mate (Ilex paraguarensis) infusions" can be a minor revision.

In the following there are some observations: 

  1. The meaning for AQL, BQL, and BEC should be given, respectively, in line 161, line 162, and line 201, where these abbreviations appear for the first time.

The meaning of these abbreviations has been added.

  1. Lines 207 - 209 – why there are not reported the percentages for Ca, Sr, and K?

We are grateful to the Reviewer for pointing this out and the percentages for Ca, Sr, and K have been reported.

  1. You say that “the research on the influence of tap water extraction on yerba mate infusions is a novelty and there is a lack of the literature data” (lines 228, 229). Correlation between the results obtained in different laboratories, from different countries, for leaching with tap water, I don’t think to be possible (it cannot be considered as being a scientific method).

We agree with the Reviewer and the following statement has been added:

Nevertheless, it would be impossible to compare the results for leaching with tap water obtained in different laboratories from different countries. Therefore it cannot be considering as being a scientific method.

  1. Same notation for two different chapters, i.e. 3.3. for “Accuracy of mineralization and extraction procedures” and for “Ultrasound–assisted extraction in comparison to a conventional brewing”.

The same notation has been fixed.

Reviewer 3 Report

This article mainly compared the extraction rate between tap water and deionized water, other parameters such as temperature, ultra-sound were applied. Manuscript was well edited, and the English is good. However, there is some place need to be explained. 

In the introduction, the importance of trace metal in the beverage needs to be dressed. Why the content of metals is important? As the author said, minerals extraction by tap water is novel, but why minerals are important to the beverage? Are minerals related to flavor or food safety?

In table 2. Do you repeat sample? What is the standard deviation of each data?

In spearman’s correlation test, it needs Xi and Yi for calculation. Please explain for example Al in total content and Al in tap water RT, what is the data used for the Xi and Yi?

Are the sample points enough? How much sample point was used for a metal element?

Please explain why only 10 elements were used in the spearman’s correlation test.

Author Response

This article mainly compared the extraction rate between tap water and deionized water, other parameters such as temperature, ultra-sound were applied. Manuscript was well edited, and the English is good. However, there is some place need to be explained. 

  1. In the introduction, the importance of trace metal in the beverage needs to be dressed. Why the content of metals is important? As the author said, minerals extraction by tap water is novel, but why minerals are important to the beverage? Are minerals related to flavor or food safety?

We agree with the Reviewer and the following paragraph has been thoroughly revised:

Their content may depends on various factors e.g. soil type, harvest season [12] and preprocessing [9]. Moreover, the extractable content of selected essentially trace and potentially toxic elements may be different in the case of infusion prepared from green and roasted yerba mate [6]. The total content of elements (in leaves and stems) is not equivalent to the extractable content (in the infusion), what was reported several times [13–15]. Therefore the determination of these elements is important evaluating the nutrition potential as well as risk assessment, however total content is not sufficient in this field.

[6]  Ribeiro, S.A.O.; da Silva, C.S.; de Araújo Nogueira, A.R.; Garcia, E.E. Solubility of Cd, Cr, Cu, Ni, and Pb and Its Correlation with Total Polyphenols and Soluble Melanoidins in Hot Infusions of Green and Roasted Mate. Biol Trace Elem Res 2022, doi:10.1007/s12011-022-03314-3.

[9]  Ramirez, M.R.; Mohamad, L.; Alarcon-Segovia, L.C.; Rintoul, I. Effect of Processing on the Nutritional Quality of Ilex Paraguariensis. Applied Sciences 2022, 12, 2487, doi:10.3390/app12052487.

[12] Camotti Bastos, M.; Cherobim, V.F.; Reissmann, C.B.; Fernandes Kaseker, J.; Gaiad, S. Yerba Mate: Nutrient Levels and Quality of the Beverage Depending on the Harvest Season. Journal of Food Composition and Analysis 2018, 69, 1–6, doi:10.1016/j.jfca.2018.01.019.

[13] Baran, A.; Gruszecka–Kosowska, A.; KoÅ‚ton, A.; Jasiewicz, C.; Piwowar, P. Content and Health Risk Assessment of Selected Elements in the Yerba Mate ( Ilex Paraguariensis , St. Hillaire). Human and Ecological Risk Assessment: An International Journal 2018, 24, 1092–1114, doi:10.1080/10807039.2017.1406304.

[14] Pozebon, D.; Dressler, V.L.; Marcelo, M.C.A.; de Oliveira, T.C.; Ferrão, M.F. Toxic and Nutrient Elements in Yerba Mate ( Ilex Paraguariensis ). Food Additives & Contaminants: Part B 2015, 8, 215–220, doi:10.1080/19393210.2015.1053420.

[15] Ulbrich, N.C.M.; do Prado, L.L.; Barbosa, J.Z.; Araujo, E.M.; Poggere, G.; Motta, A.C.V.; Prior, S.A.; Magri, E.; Young, S.D.; Broadley, M.R. Multi–Elemental Analysis and Health Risk Assessment of Commercial Yerba Mate from Brazil. Biol Trace Elem Res 2022, 200, 1455–1463, doi:10.1007/s12011–021–02736–9.

  1. In table 2. Do you repeat sample? What is the standard deviation of each data?

Each samples was measured triple. Moreover, the complete analytical process was repeated twice for each digest and extracts. All results in Table 2 have been presented as mean±SD according to the Reviewer’s comment.

  1. In spearman’s correlation test, it needs Xi and Yi for calculation. Please explain for example Al in total content and Al in tap water RT, what is the data used for the Xi and Yi?

In spearman’s correlation test, Xi and Yi were respectively the total and extractable contents of the same element. In mentioned example, Xi was total content of Al and Yi was the content leached by tap water at room temperature.

  1. Are the sample points enough? How much sample point was used for a metal element?

In our opinion, 7 samples are enough. The Spearman’s rank correlation, as a non-parametric test, is more resistant to outliers than the Pearson correlation coefficient. Therefore, showing a normal distribution, we considered to apply it either.

  1. Please explain why only 10 elements were used in the spearman’s correlation test.

We decided to use only these elements which were detected above quantification limits in each sample. We took this as a necessary parameter due to small number of samples (n=7). For this reason, some elements which were determined most of sample were rejected (e.g. Cr and Co). The statement has been added in the paragraph.

Due to the normality of the data distribution was rejected as well as small number of samples (n=7), the Spearman’s rank correlation coefficient (rs) was applied (Figure 1), presenting the pairwise associations between total content and extraction procedures (A–D), Furthermore, 10 elements were only selected (which were determined AQL in each digest and extract).

Reviewer 4 Report

The main purpose of this work was to evaluate the effect of extraction way on the concentrations of seven pure yerba mate (Ilex paraguariensis) infusions.

In general, this manuscript presented relevant results about the effect extraction processing of seven pure yerba mate infusions. So, the main question addressed in this article is the extraction process of yerba mate.

There are some specific comments listed below:

Page 3, Table 1:

What is the difference about samples two and seven? Maybe type (con palo) or packing type (paper pack), please verify.

There is no discussion about information presented in this Table 1.

For example, what is the importance of evaluating the packaging? It is not clear, since it only mentions the use of two packages, paper, and plastic.

There is no discussion too about type of yerba “con palo” or “despalada” in obtained results.

Similarly, was there any difference between the yerba extraction parameters for the studied countries, Brazil, Paraguay, and Argentina?

Since the use of tap water is part of one of the studied process methodologies, it would be better to insert a characterization analysis of this water.

Page 10, lines 341-342

It is not clear what extraction procedure was reported before. Since, I suggest include reference as follow: “Above results may be easily compared with hot water extraction procedures reported by Baran et al. (2018).”

The conclusions are consistent with the evidence and arguments presented in this study.

Author Response

The main purpose of this work was to evaluate the effect of extraction way on the concentrations of seven pure yerba mate (Ilex paraguariensis) infusions.

In general, this manuscript presented relevant results about the effect extraction processing of seven pure yerba mate infusions. So, the main question addressed in this article is the extraction process of yerba mate.

There are some specific comments listed below:

  1. Page 3, Table 1: What is the difference about samples two and seven? Maybe type (con palo) or packing type (paper pack), please verify.

According to the Reviewer’s comment, we decided to define yerba mate type, which is related to different preprocessing procedure (Table 1). Moreover, the following statement has been added in the text: All products were pure (100%) and roasted yerba mate (except of sample 2).

  1. There is no discussion about information presented in this Table 1.

We agree with the Reviewer and the following paragraph has been revised.

Moreover, the order K > Ca > Mg > Mn > P > S > Al > Fe > other elements, was observed for most of samples (except sample 1 where S > P). This observation was also reported in several studies [8–10, 12, 15]. In the case of other elements (ranging 20–100 mg kg–1), the following orders (Na > Zn and Rb > Sr) were generally observed (sample 1, 2, 6, 7). It is worth noting that the only green yerba mate (sample 2) was an exceptional case (in the whole series) where Rb > Zn and Ni > Se were observed. In turn, different pair (Zn > Na and Sr > Rb) were noticed for samples 3 and 4 which were the same brand. Additionally, sample 5 (Argentinian mate con palo) was an exception for which the following series was noted (Na > Zn and Sr > Rb).

  1. For example, what is the importance of evaluating the packaging? It is not clear, since it only mentions the use of two packages, paper, and plastic.

The following statement has been added.

It is worth mentioning that higher sulfur content was found in samples 1, 2 and 7. While all samples are Polish brand products sold in plastic bags, it is difficult to clearly determine the reason.

  1. There is no discussion too about type of yerba “con palo” or “despalada” in obtained results.

The following statement has been added.

Nevertheless, other characteristic orders (in the total content) were not observed comparing the type and country of origin, which is associated with too few samples (n=7).

  1. Similarly, was there any difference between the yerba extraction parameters for the studied countries, Brazil, Paraguay, and Argentina?

The following paragraph has been added.

No significant differences were observed between different countries of origin, types and packing material comparing medians of extraction percentages. It is surprising that samples 2 and 7, which both were Brazilian mate (despalada type) of the same brand, had the noticeable differences in the extractable content of various elements. The higher extraction percentages for Fe, Cu and Zn were observed for sample 2 (green mate) had, while K, Mg, Na, Ni, P, Rb and S for sample 7 (roasted mate). Moreover, Co and Se were detected AQL in extracts of sample 7. Similar percentages were found in the case of Al, As, Ca, Mn, Sr, while Cr and Cd were not extractable in both samples. The explanation is probably a thermal degradation of polyphenols during preprocesses (e.g. drying and roasting). The following influence on the extraction percentages were reported suggesting that these elements (which the content increased) were bonded to the polyphenols [6].

[6]  Ribeiro, S.A.O.; da Silva, C.S.; de Araújo Nogueira, A.R.; Garcia, E.E. Solubility of Cd, Cr, Cu, Ni, and Pb and Its Correlation with Total Polyphenols and Soluble Melanoidins in Hot Infusions of Green and Roasted Mate. Biol Trace Elem Res 2022, doi:10.1007/s12011-022-03314-3.

  1. Since the use of tap water is part of one of the studied process methodologies, it would be better to insert a characterization analysis of this water.

The characterization of tap water was reported in Supplementary Materials (Table S1).

  1. Page 10, lines 341-342. It is not clear what extraction procedure was reported before. Since, I suggest include reference as follow: “Above results may be easily compared with hot water extraction procedures reported by Baran et al. (2018).”

The statement has been changed according to the Reviewer’s comment.

  1. The conclusions are consistent with the evidence and arguments presented in this study.

We sincerely thank the Reviewer for the positive reception of this section.

Round 2

Reviewer 1 Report

The authors fairly responded to the comments and the manuscript improved after revision.